# The Effects of *Houttuynia cordata* Thunb and *Piper ribesioides* Wall Extracts on Breast Carcinoma Cell Proliferation, Migration, Invasion and Apoptosis

**DOI:** 10.3390/molecules25051196

**Published:** 2020-03-06

**Authors:** Subhawat Subhawa, Teera Chewonarin, Ratana Banjerdpongchai

**Affiliations:** Department of Biochemistry, Faculty of Medicine, Chiang Mai University, Chiang Mai 50200, Thailand; subhawat_s@cmu.ac.th (S.S.); teera.c@cmu.ac.th (T.C.)

**Keywords:** *Houttuynia cordata* Thunb, *Piper ribesioides* Wall, extracts, anti-cancer, anti-proliferation, breast cancer, cancer progression, migration, invasion

## Abstract

*Houttuynia cordata* Thunb. (HCT) and *Piper ribesioides* Wall. (PR) are common herbs that are widely distributed throughout East Asia and possess various biological properties including anti-cancer effects. However, in breast cancer, their mechanisms responsible for anti-carcinogenic effects have not been clarified yet. In this study, the inhibitory effects of HCT and PR ethanolic extracts on breast cancer cell proliferation, migration, invasion and apoptosis were examined. In MCF-7 and MDA-MB-231 cells, HCT and PR extracts at low concentrations can inhibit colony formation and induce G1 cell cycle arrest by downregulating cyclinD1 and CDK4 expression. Additionally, HCT and PR extracts also decreased the migration and invasion of both breast cancer cell lines through inhibition of MMP-2 and MMP-9 secretion. Moreover, the induction of apoptosis was observed in breast cancer cells treated with high concentrations of HCT and PR extracts. Not only stimulated caspases activity, but HCT and PR extracts also upregulated the expression of caspases and pro-apoptotic Bcl-2 family proteins in breast cancer cells. Altogether, these findings provide the rationale to further investigate the potential actions of HCT and PR extracts against breast cancer in vivo.

## 1. Introduction

Global cancer statistics from 2019 report that breast cancer is the most frequently diagnosed cancer and the leading cause of cancer death among females worldwide [1]. There are many signaling pathways related to the initiation of carcinogenesis, the reproductive maintenance systems and their involvement in cells, through the highly proliferative cells during tumorigenesis. Moreover, these are related to the metastasis of cancer cells and can be caused by violence in breast cancer cells. Despite significant advances in the diagnosis and treatment of breast cancer, several major unresolved clinical and scientific problems remain, for example: Prevention; tumor progression; recurrence; and metastasis or treatment [2]. Moreover, breast cancer cells can metastasize anywhere in the body, although they mainly metastasize to bones, lungs, regional lymph nodes, the liver and the brain, with the most common site being bones [3]. The metastasis mechanism contains many subsequent steps. First, cancer cells begin to invade from the primary tumor site and migrate intravasate into the blood or lymphatic vessels. Second, metastatic cancer cells change some phenotypes that lead to the downregulation of cell-cell adhesion molecules, such as *E*-cadherins, while concurrently upregulating the expression of mesenchymal markers such as *N*-cadherin and activating metalloproteinases (MMPs). Third, the extracellular matrix (ECM) is degraded mainly through MMPs such as MMP-2 or MMP-9. Invasion is preceded by the degradation of the ECM to enable the infiltration of cancer cells [4]. The knowledge of its mechanisms is still fragmentary and must be summarized to improve our therapeutic approach and influence the long-term and effective control of breast cancer progression.

Clinically, breast cancer can be divided into distinct subtypes, including prognostic, which can be classified by severity [5]. There are many breast cancer treatments, such as chemotherapy, radiotherapy, and targeted therapy. The mechanisms of these treatments can modulate pro- and anti-apoptotic proteins by upregulating the expression of pro-apoptotic proteins, such as p53-effector related to p21, BAX and Noxa. Conversely, they can also lead to the downregulation of anti-apoptotic proteins [6]. Yet these methods, like all those using drugs, will result in side effects. Currently, herbal medicines play a role in the prevention of breast cancer and have fewer side effects than using chemotherapeutic drugs since herbs are plants that most people take orally as a dietary supplement. Therefore, the development of herbal extracts to be used as a dietary supplement to prevent cancer and treat breast cancer is required.

*Houttuynia cordata* Thunb and *Piper ribesioides* Wall are widespread herbs found in Northern Thailand. There is a long history of herbal medicine plants widely distributed in East Asia, with Chinese people having utilized herbs and plants to treat various diseases for a long time. Moreover, they are intriguing natural products which are widely used as food supplements and to promote health [7]. Furthermore, *H. cordata* possesses anti-cancer, anti-diabetics, and anti-inflammation properties. However, for *P. ribesioides*, there has been little investigation or research conducted regarding such features. *P. ribesioides* contains various phytochemicals, including camphene, sabinene, and β-caryophyllene [8]. This study aimed to study and investigate the effects of anti-proliferation, anti-invasion, anti-migration and apoptosis induction of both plants ethanolic extracts on two different breast cancer cell types, including MCF-7 (non-invasive breast cancer cell) and MDA-MB-231 (invasive breast cancer cell). Based on the potential actions on breast cancer cells, both extracts can be developed as anti-cancer agents in order to prolong life among breast cancer patients. However, the anti-carcinogenic activity against the breast cancer and toxicity tests of both extracts need to be verified using animal models and clinical trials.

## 2. Results

### 2.1. Identification of Phytochemical Compositions in H. cordata and P. ribesioides Extracts

Many phytochemical compounds in both plants have been reported, such as phenolic acids and alkaloids [7,8]. We investigated the total phenolic acid contents, flavonoids and antioxidant activity, as shown in Table 1. The phenolic acid compositions and flavonoids in *P. ribesioides* were higher than *H. cordata*. Moreover, both extracts from plants possessed radical scavenging activity by using DPPH assay. The values of IC50 of *H. cordata* and *P. ribesioides* were 234.6 ± 11.9 and 153.8 ± 4.4 when compared to vitamin C (Table 1). Additionally, the HPLC chromatography exhibited related results to the total phenolic content, total flavonoids, and DPPH-radical scavenging activity. To determine and quantify phenolic acids and flavonoids, both ethanolic *Houttuynia cordata* Thunb. (HCT) and *Piper ribesioides* Wall. (PR) extracts were analyzed using the standard curve compared to 11 phenolic acids and flavonoids standards. Six phenolic acids were compared as gallic, vanilic, ferulic, *p*-coumaric, chlorogenic and rosmarinic acids; and five flavonoids were compared as catechin, rutin, quercetin, apigenin, and luteolin. Eleven phenolic acids and flavonoids in both extracts were identified according to HPLC analysis (Table 2 and Figure 1), in which chlorogenic acid and rutin were predominant phenolic acids and flavonoids in HCT extract, respectively. For PR extract, the predominant phenolic acid and flavonoids were unidentifiable due to the limitations of the standards. However, the researchers had investigated more active compounds compared to alkaloid derivatives, such as piperine by HPLC analysis. We found that piperine was the predominant compound in ethanolic PR extract (Table 2 and Figure 1). Further studies had been characterized for volatile substances in both ethanolic plant extracts by using the GC-MS (Gas chromatography–mass spectrometry) method as shown in Figure 2A,B. The HCT extract contained 9.42% of decanoic acid, 10.5% of *n*-hexadecanoic acid, 11.67% of 9,12-octadecatrienoic acid and 14.74% of cholest-5-en-3-ol (Table 3). For PR extract, the major compound was piperine (49.51%), as shown in Table 4. The researchers, therefore, studied the anti-cancer potential effect of both ethanolic plant extracts in further experiments.

### 2.2. Cytotoxic Effects of H. cordata and P. ribesioides Extracts on Breast Cancer Cells, PBMCs and MCF-10A

HCT was cytotoxic against many types of cancer cells, for example MCF-7 [9] and MDA-MB-231 [10]. Yet, there is still no report of the PR toxic effect on both types of breast cancer cells. To investigate the effects on cell viability, both HCT and PR extracts were prepared in various concentrations (100–500 µg/mL) and were incubated in both MCF-7, MDA-MB-231, peripheral blood mononuclear cells (PBMCs), and MCF-10A (human breast epithelial cell line) for 24, 48, and 72 h. In addition, IC20 and IC50 values (inhibitory of concentration at 20% and 50%) were evaluated by using results of cell viability. As shown in Figure 3, both ethanolic extracts decreased the viability of both breast cancer cells, but not PBMCs, in dose- and time-dependent manners. Although the viability of human mammary epithelial cell line MCF10A tends to decrease when treated with increasing doses of HCT and PR extracts (Figure 3), IC20 and IC50 of both extracts in normal cells MCF10A were higher than in breast cancer cells MCF-7 and MDA-MB-231 (Table 5). These findings indicate that breast cancer cells were more sensitive to ethanolic HCT and PR extracts than normal mammalian cells. Furthermore, IC20 and IC50 of PR extracts in breast cancer cells were lower than in HCT extract (Table 5), implying that both cancer cells were more sensitive to PR extract. Besides, the results also show that the viability of the cells treated with both ethanolic extracts at 24 h was significantly different from at 48 and 72 h (Figure 3). However, there were no significant differences between breast cancer cell viability at 48 and 72 h of incubation (Figure 3).

### 2.3. Effects of H. cordata and P. ribesioides Extracts on Growth Inhibition in Breast Cancer Cells

*H. cordata* plays an important role in cell proliferation and cell cycle progression [9,11]. However, the anti-proliferation effect of *P. ribesioides* has not yet been studied. This anti-proliferation with both *H. cordata* and *P. ribesioides* extracts in both breast cancer cells was verified. The cells were treated with various concentrations of HCT and PR, at concentrations not more than IC20, to prevent cell death from such concentrations. The colony formation assay was measured, and HCT and PR dose-dependently inhibited colony formation in both MCF-7 and MDA-MB-231 cells (Figure 4A). MCF-7 was more sensitive to both HCT and PR compared to MDA-MB-231 cells. However, the low concentrations of HCT extract were more effective than in PR extract, since the cell cycle is disrupted and involved in the proliferation of cancer cells in carcinogenesis [12]. Consequently, the cell cycle assay was verified to demonstrate the proliferation-inhibitory effects of HCT and PR (Figure 4B). After treatment with both ethanolic extracts, MCF-7 cells were arrested at G1 stages in a dose-dependent manner and significantly increased in the G1 phase. For MDA-MB-231 cells, only PR treatment at 200 µg/mL significantly increased in the G1 phase. These results suggest that both HCT and PR inhibited the proliferation of both breast cancer cells by cell cycle arrest (Figure 4A). To confirm that HCT and PR-induced G1 arrest in breast cancer cells, both breast cancer cells were treated with HCT and PR extracts with the same condition for 24 hours’ treatment, and cell cycle-related proteins such as p21, cyclin D1 and CDK4 [13] were detected. The results show that cyclin D1 and CDK4 decreased in both MCF-7 and MDA-MB-231 cells and correlated with the results from the colony formation assay and cell cycle analysis. It was also found that there was an increase of p21 in MCF-7, but in MDA-MB-231, p21 was not detectable (Figure 5). These findings show that both ethanolic HCT and PR extracts inhibited cell growth and arrested the G1 phase (Figure 4B), by decreasing the expression of cyclin D1 and CDK4 and altering the p21 expression level, as shown in Figure 5.

### 2.4. Effect of H. cordata and P. ribesioides on Breast Cancer Migration and Invasion

Cancer cell migration and invasion are very important characteristics of breast cancer cells at the metastasis steps. Cancer cells secrete metalloproteinases (MMPs), such as MMP-9, to degrade the extracellular matrix [2]. Moreover, in both *H. cordata* and *P. ribesioides,* no research studies have previously been undertaken on the inhibition of migration and invasion in breast cancer cells. To explore the effect of both *H. cordata* and *P. ribesioides* on cancer cells migration and invasion, both MCF-7 and MDA-MB-231 cells were treated with both *H. cordata* and *P. ribesioides* at non-toxic concentrations in incomplete media. Both HCT and PR significantly inhibited both MCF-7 and MDA-MB-231 migration in a dose-dependent response, compared with the control using the wound healing assay (Figure 6). The researchers found that both ethanolic extracts decreased MMP-2 and MMP-9 secretions in MDA-MB-231 (Figure 7A). Meanwhile, both ethanolic extracts in the same concentrations inhibited the invasion of MDA-MB-231 cell by using a Transwell-invasion assay (Figure 7B). HCT inhibited the breast cancer cells migration to a better degree than PR. These results indicated that both HCT and PR extracts inhibited breast cancer cell migration and invasion though the inhibition of MMPs secretion. 

### 2.5. Effects of H. cordata and P. ribesioides Extracts on Apoptosis Induction in Breast Cancer Cells

Various reports have investigated HCT extracts-induced apoptosis in many types of cancer, such as human primary colorectal cancer cells [14]. Nevertheless, for PR there is no study on apoptosis induction. However, there are reports about anti-cancer effect of other *Piper* species [15]. Furthermore, the relationship of doses and types of cell death remains intriguing, for example, in low doses it could produce apoptosis, and at high dose the necroptosis are stimulated by, e.g., shikonin [16]. To further investigate apoptosis induction by both ethanolic HCT and PR extracts, the apoptosis cell death was evaluated using annexin V-fluorescein isothiocyanate (FITC)/propidium iodide (PI) staining and analyzed by flow cytometry. Each IC20 and IC50 of the ethanolic HCT and PR extracts were treated in both breast cancer cells, and the cells were then induced to undergo apoptosis. The percentage of apoptotic cells was significantly increased in both cancer cells in concentration-dependent manners (Figure 8). Compared with the concentrations of the extracts, the MCF-7 cell line was more sensitive than MDA-MB-231.

In addition, the activity of caspases was determined to investigate caspase-dependent apoptosis cell death by using a colorimetric assay kit. Caspase activities are used to demonstrate the pathways of the apoptotic cell death pathway [17]. Both cell types were treated with the same conditions as in the experiments of flow cytometry, or caspase activity was assayed by using a spectrophotometric microplate reader. The caspases -3, -8, and -9 activities significantly increased in concentration-dependent manners (Figure 9) compared to the untreated control cells, and they correlated with the results from annexin V-FITC/PI by flow cytometry. However, caspases-3 is not detectable in the MCF-7 cell line, corresponding to a previous study [18]. The results indicate that both ethanolic extracts induced apoptosis in MCF-7 and MDA-MB-231 via both extrinsic and intrinsic pathways. To confirm HCT and PR induced apoptosis, apoptotic-protein levels in Bcl-2 family proteins were investigated by Western blotting. When both breast cancer cells were treated with the IC20 and IC50 concentrations of each ethanolic extract, the results show that the anti-apoptotic protein Bcl-xl significantly decreased, whereas the pro-apoptotic proteins significantly increased in dose-dependent manners (Figure 10). These results indicate that both ethanolic extracts induced apoptosis using cytotoxic doses in both breast cancer cells. Caspase-8 is the marker of the extrinsic pathway, whereas the caspase-9 is responsible in the intrinsic pathway [19,20]. Yet, these two caspases act as if they were at the initiation caspases [21]. Caspases-3, -6, and -7 are the effector caspases or executioner caspases that are activated in the cascade of both intrinsic (mitochondrial) and extrinsic (death receptor) pathways [22]. MCF-7 cells do not possess caspase-3 proteins. Hence, to confirm the effect of apoptosis at the molecular pathways, the apoptosis-mediated proteins were determined. The Bcl-2 family consists of three main types: anti-apoptotic (such as Bcl-xl); pro-apoptotic proteins with multiple Bcl-2 homology domains (such as BAX); and BH3-only proteins (such as Noxa, Bid) [23,24]. The immunoblotting of caspase-7, -9, -8, Bid/tBid, BAX, and Noxa expression level was measured. It was found that the expression of Bcl-xl decreased, whereas the Bid/tBid protein levels were modulated as a decrease of the pro-form and an increase of the truncated Bid (tBid), confirming the involvement of the extrinsic pathway as the enhancement of caspase-8 activity in MCF-7. For MDA-MB-231 cells, the caspase-3 was activated together with caspase-8 and 9 in both the caspases activities, and corresponds with the caspase protein expression levels. The Bcl-2 protein expression levels followed the same pattern in accordance with those of MCF-7 (Figure 10).

## 3. Discussion

The classification of breast cancers is associated with the hormone responsive receptors in the cancer cells. The presence or absence of the receptors in the cancer cells determines the invasiveness and the aggressiveness of breast cancers [25]. A hormone receptor-positive tumor is the breast cancer that has theses following receptors, estrogen receptor (ER), progesterone receptor (PR), and epidermal growth factor receptor (HER-2/Neu). On the other hand, triple-negative breast cancer (TNBC) refers to cancer that has a complete lack of those aforementioned receptors, and this type of cancer tends to possess more aggressive phenotypes [26]. In this study, the mammary epithelial cancer cell line MCF-7 (estrogen receptor-positive cancer type) and human adenocarcinoma cell line MDA-MB-231 (TNBC type) were selected to represent the non-aggressive and highly aggressive breast cancer, respectively. Both cell lines were used for determining the inhibitory effects of HCT and PR extracts on the cell-proliferation, migration, and invasion, as well as apoptosis. Compared to MCF-7 cells, the studies about the anti-breast cancer effects of herbal extracts on MDA-MB-231 cells were limited. 

Previous studies reported that HCT contains a range of phenolic and flavonoid compounds, such as chlorogenic acid, hyperin, quercitrin, rutin, and quercetin [27,28]. Using HPLC chromatography, in the present study, the researcher found that chlorogenic acid is the most abundant phenolic compound in the ethanolic extract of HCT. Chlorogenic acid (CGA) has been revealed to possess anti-cancer potentials on breast cancer. Through the inhibition of protein kinase C signaling, CGA induces cell cycle arrest and apoptosis in MCF-7 and MDA-MB-231 cells [29], which is consistent with effects of HCT extract in this study. Thus, CGA is a promising bioactive phenolic compound of HCT extract on cell cycle arrest and apoptosis induction in breast cancer cell lines. For *Piper* spp. (Piperaceae), such as *P. nigrum* and *P. tuberculatum,* piperine is a standard alkaloid that is used to measure the efficiency of solvent extraction for the plants [30,31]. Besides, there have also been reports about the effect of piperine against breast cancer. Piperine suppresses the proliferation and migration of triple negative breast cancer cell lines by reducing the levels of MMP-2 secretion [32,33]. In agreement with these studies, PR extract, which contains a high amount of piperine, also decreased the colony formation and migration of aggressive human breast cancer cell MDA-MB-231. Therefore, the researchers suggest that piperine is likely to be the potential active compound that is responsible for the anti-cancer bioactivities of PR extract. 

Apart from HPLC, both plant extracts were subjected to GC-MS analysis for identifying the phytochemical compositions including alkaloids and some volatile components, in which the HPLC system could not be determined. The lipophilic phytochemicals in HCT extract consist of alpha-tocopherol, decanoic acid, *n*-hexadecanoic acid and 9,12-octadecatrienoic acid, which are all comparable to the results from previous studies [27,34,35,36]. Although camphene, sabinene, and β-caryophyllene are common substances found in piperine plants [8], the researchers found that piperine, *n*-hexadecanoic acid and cis-vaccenic acid were mainly presented in the PR extract. This might be due to the different condition used in GC-MS analysis. Decanoic acid, a straight-chain saturated fatty acid, has been previously reported as the antibacterial- and the anti-inflammatory agent [36]. The derivatives or conjugated forms of decanoic acid were also found as contributing agents, supporting the anti-cancer agents, a class of cell-permeable Bcl-2 binding peptides, by chemically attaching to a decanoic acid as the cell-permeable moiety [37]. Besides, *n*-hexadecanoic acid that was found in both extracts has been reported for its activity on apoptosis induction in breast cancer [38]. Regarding the phytochemical studies, it is possible that decanoic acid and its derivatives are the active compounds of HCT and PR extracts on apoptosis induction. 

Low concentrations (less than IC 20) of ethanolic HCT and PR extracts were found to significantly inhibit cell growth and colony formation in both breast cancer cell types. These concentrations also significantly induced the cell cycle arrest at the G1 phase, and the results were confirmed with cell cycle progression histograms and the expression of cell cycle-related proteins by Western blot analysis. The cyclin D1 and CDK4 proteins, which are cell cycle checkpoint proteins for the G1 phase, were found to be significantly decreased. These resulted in the inhibition of colony formation and cell cycle arrest [39]. The p21 protein expression levels in MCF-7 and MDA-MB-231 cell lines were different, because MDA-MB-231 is often accompanied by a higher frequency of p53 gene mutations [40], and there is also no evidence of p21 protein expression in such invasive cells [41,42]. These results indicate that HCT and PR extracts induced G1-arrest in MCF-7 cells, which depended on p21 expression, but not in the case of MDA-MB-231 cells. However, we still observed that the G1 arrest occurred in MDA-MB-23. This result might be due to the activation of a p-21-independent pathway, such as the EGFR signaling pathway [43]. There are several previous reports about HCT-induced cell cycle arrests in various types of cancer, including lung cancer cell (A549 cell), [11], hepato-cellular carcinoma (HepG2 cell) [44], and also breast cancer cells (MCF-7 cell) [9]. The inhibition of DNA synthesis, cell cycle arrest, and the inhibition of proliferation are associated with increased p21(Waf1/Cip1) expression [45].

Breast cancer progression requires multiple steps, including migration, invasion, angiogenesis, and metastasis [2]. In particular, the anti-migration and invasion effects of HCT and PR extracts have not yet been studied in either MCF-7 or MDA-MB-231 cells. Matrix metalloproteinases (MMPs) are the proteolytic enzymes that are important for cancer cell invasion. MMPs are capable of degrading a range of extracellular matrix proteins, then allowing the cancer cells to migrate and invade, away from the primary site [46]. Among various subtypes of MMP proteins, MMP-2 and MMP-9 are frequently found to be related with cancer cell invasion and aggressiveness. MMP-2 and MMP-9 have been reported as proteases for the degradation of type IV collagen, a major component of basement membrane [47,48]. Kim et al. reported that MMP-2 secretion in MCF-7 is extremely low compared to MDA-MB-231 [49]. Consistently, the results from the present study show that MMP-2 and -9 activities in MCF-7 cell lines were not detectable (data not shown). Thus, MDA-MB-231 cells were selected for investigating the mechanisms underlying the anti-migration and invasion properties of HCT and PR extracts. To avoid the influence from cytotoxic effects of the extracts, the migration and invasion potentials of MDA-MB-231 cells were examined in the serum-free media condition. The results indicate that HCT and PR extracts at non-toxic doses could inhibit cancer cell migration and invasion by decreasing the MMP-2 and -9 secretion. However, the underlying mechanisms related to the decreased secretion of MMPs should be further determined.

Beyond the anti-migration and anti-invasion effects, HCT and PR extracts could induce apoptosis in both MCF-7 and MDA-MB-231 cells via the caspase-dependent mechanism. Regarding the apoptosis pathways, the extrinsic pathway, caspase-8, is the initiator caspases that activates downstream caspases, including caspase-3, -7 [50]. Caspase-9 is activated by internal signaling, increasing Noxa and BAX protein, leading to the interaction and inhibition function of Bcl-xl protein [51,52]. The active caspase-8 in the cross-talk apoptosis pathway is activated via the induction of caspase-8 to cleave Bid and become truncated Bid (tBid) [53,54]. Both caspases then activated caspase-3 or -7 in a downstream cascade and the cells underwent apoptosis [55,56]. Although there is a report about the effector caspase between caspase-3 and caspase-7 [57], caspase-3 and caspase-7 play separate roles during apoptosis [22,58]. Thus, the researchers focused on the effects of both extracts on the aforementioned caspases types. Western blotting analyses confirm that both HCT and PR extracts enhanced the expression of caspase-7, -8 and -9 at protein levels. Moreover, caspase-3, -8 and -9 activities were inhibited in MDA-MB-231 cells treated with high doses of HCT and PR extracts. Similar results excluding caspase-3 activity were observed in MCF-7, according to the lack of caspase 3 in MCF-7 cells [19].

Taken together, our results demonstrate the novel mechanisms associated with the inhibitory effects of HCT and PR extracts on breast cancer cell growth, migration, and invasion. Both extracts at sub-cytotoxic doses induced cell cycle arrest via the modulation of cyclin D1, CDK4 and p21 protein expressions. Besides, both extracts suppress breast cancer cell migration and invasion by inhibiting MMPs secretion. Furthermore, the toxic concentrations of both extracts (IC_20_ and IC_50_) can induce breast cancer cell apoptosis through the upregulated expression of caspases and pro-apoptotic proteins, and the decreased expression of anti-apoptotic proteins. To enhance the value of the plants, anti-carcinogenic effects of HCT and PR extracts against breast cancer should be further verified in the animal model.

## 4. Materials and Methods

### 4.1. Reagents

Dulbecco’s Modified Eagle Medium—high glucose (DMEM-HG) (12800-58), Roswell Park Memorial Institute (RPMI)-1640, DMEM/Ham’s F-12 (GIBCO-Invitrogen, Carlsbad, CA, USA), fetal bovine serum (FBS), phosphate-buffered saline (PBS)and trypsin-EDTA solution were purchased from Gibco (Grand Island, NY, USA) Dimethyl sulfoxide (DMSO), and sulforhodamine B (SRB) was purchased from Sigma Chemical, Inc. (St Louis, MO, USA) The substrate of caspase-9 (LEHD-*para*-nitroaniline; LEHD-*p*-NA), caspase-8 (IETD-*para*-nitroaniline; IETD-*p*-NA), caspase-3 (DEVD-*para*-nitroaniline; DEVD-*p*-NA), and SuperSignal West Pico Chemiluminescent Substrate were obtained from Invitrogen (Thermo Fisher Scientific Inc., Waltham, MA, USA). Primary antibodies against caspase-9 (ab32539), caspase-8 (ab25901), caspase-7 (ab25900), BAX (ab32503), Bcl-xl (ab32370), Bid (ab2388), Noxa (ab13654), actin (ab8227) and peroxidase-labeled secondary antibodies; anti-rabbit IgG (ab97051), anti-mouse IgG (ab97046) were purchased from Abcam (Cambridge, UK). Protease inhibitor was obtained from Roche Diagnostics, Mannheim, Germany.

### 4.2. Plant Sample Preparation

The leaves and stems of *H. cordata* (HCT) and *P. ribesioides* (PR) were provided from Prolac (Thailand) Corporation, Ltd., Lamphun, Thailand. One-hundred grams of each plant powder were roughly blended and then extracted in 1:10 ratios with 80% ethanol by stirring overnight. The supernatant from extraction was directly filtered through Whatman filter paper No.1, followed by evaporation using a rotating evaporator (40 °C) and pressure at 100–150 mbar. After that, the fraction was lyophilized.

### 4.3. Measurement of Total Phenolic Compounds

The total phenolic compounds in these two plant extracts were determined by using a Folin–Ciocalteu assay [59]. In total, 20 µL of each extract was mixed in 96 well plates with 100 µL of Folin-Ciocalteu reagent and 80 µL of sodium carbonate (Na_2_CO_3_) (75 g/L) solution. The reaction mixture was incubated at room temperature for 30 min and then the absorbance was measured at 765 nm. A standard curve was attained using various concentrations of standard gallic acid. The total phenolic content was expressed as milligrams of gallic acid equivalents (GAE) per gram dry weight of the extracts.

### 4.4. Determination of Total Flavonoid Compound

The total flavonoid compound in HCT and PR was determined by using an aluminum chloride colorimetric assay. Various diluted solutions of 25 µL were mixed with 125 µL distilled water and 7 µL of 5% sodium nitrite (NaNO_2_) in 96-well plates. The reaction mixture was incubated at room temperature for 6 min. Then, 15 µL of 10% aluminum chloride (AlCl_3_) was added to the reaction mixture and incubated for 5 min at room temperature. Following that, 500 µL of 1 M NaOH and 27.5 µL of distilled water were added. The absorbance was measured at 510 nm. The total flavonoid content was calculated by catechin standard curve and expressed as milligrams of catechin equivalent (CE) per gram weight of both plant extracts [60].

### 4.5. Determination of Anti-OxidantActivity by 2,2-Diphenyl-1-Picrylhydrazyl (DPPH) Assay

The free radical scavenging activity of HCT and PR was determined by DPPH radical scavenging assay [61]. Both plant extracts were dissolved in DMSO at 100 µg/mL. After that, 20 µL of both plant extracts in various diluted solutions were mixed with 180 µL of 0.2 mM DPPH (2,2-diphenyl-1-picrylhydrazyl) in 96 well-plates and incubated for 30 min at room temperature in the dark. The absorbance of DPPH was measured at 517 nm. Mixing methanol with DMSO was used as a blank control, and the scavenging activity of each plant extract was compared with the vitamin C standard curve. IC50 of each extract was calculated and compared with the vitamin C standard curve.

### 4.6. Determination of Phenolic Compounds by Reversed Phase—High Performance Liquid Chromatography (HPLC)

This investigation was determined by HPLC using the column Phenomenex RP-Gemini NX C18 (250 mm × 4.6 mm, 5 μm), and diameter was of particle size. The mobile phases were 0.1% trifluoroacetic acid (TFA): water as solvent A and methanol as solvent B, at a flow rate of 1 mL/min, and both were in the gradient elution program. Ten microliters of the samples were injected into the column with a flow rate of 1.0 mL/min. Standards were monitored at 280 and 293 nm, respectively. The contents of each phenolic compound were calculated by the HPLC peak area under the curve, compared with the standard calibration curve [62].

### 4.7. Characterization of Phytochemicals by Using GC-MS Technique

A GC-MS analysis was carried out, using Agilent Technology GC 7890A coupled to Agilent Technology MSD 5975C (EI) (Agilent, Santa Clara, CA, USA). Both plant extracts were performed on a DB-5MS column fused with silica of 30 m × 0.25 mm ID × 0.25 μm film thickness. The oven temperature was programmed from 80 °C at 10 °C/min, to 200 °C at 12 °C/min, to 260 °C (30 min). Helium gas (99.999%) was used as the carrier gas at a constant flow rate of 1mL/min, and an injection volume of 1 μL was employed (split ratio of 10:1) at an injector temperature of 250 °C; the ion-source temperature was set at 280 °C. The compounds were detected in the range 50–550 amu [63]. The molecular weight and structure of the compounds of the test materials were ascertained by interpretation of the mass spectrum of the GC­MS, using the database of the National Institute of Standards and Technology (NIST).

### 4.8. Cell Culture

Human breast cancer cell lines (MDA-MB-231 and MCF-7) were generous gifts from Professor Dr. Prachya Kongtawelert, Excellence Center of Tissue Engineering and Stem Cells, Department of Biochemistry, Faculty of Medicine, Chiang Mai University. Both cells were cultured in Dulbecco’s Modified Eagle Medium with 25 mM NaHCO_3_, 100 Units/mL penicillin, and 100 µg/mL streptomycin, and supplemented with 10% heat-inactivated fetal bovine serum, grown at 37 °C under a 5% CO_2_ atmosphere. The cells were harvested, plated or sub-cultured when they obtained a 70–80% confluence for preservation or cycle passages.

A buffy coat from volunteers at the Blood Bank Unit was separated by histopaque-1077, following density gradient centrifugation standard protocol to obtain PBMCs. The PBMCs were cultured in RPMI-1640 medium and supplemented with 10% FBS, 2 mM glutamine, 100 Units/mL penicillin, and 100 μg/mL streptomycin at 37 °C in a 5% CO_2_ atmosphere. The informed consent was signed, and the approval was available from the Ethic Committee of the Maharaj Nakorn Chiang Mai Hospital, Faculty of Medicine, Chiang Mai University (no. EXEMPTION-6146/2562, approved on 4 November 2019).

The human MCF10A mammary epithelial cell line was purchased from ATCC and cultured in DMEM/Ham’s F-12 supplemented with 5% HS, 20 ng/mL EGF, 0.5 mg/mL hydrocortisone, 100 ng/mL cholera toxin, 10 µg/mL insulin, 50 U/mL penicillin and 50 µg/mL streptomycin.

### 4.9. Cell Viability and Proliferation Assay

The procedure of SRB assay was performed by seeding the cells in a 96-well culture plate (10,000 cells/well) in 100 µL complete medium. The herbal extracts treatment conditions were prepared in 100 µL complete medium, and the solution was added to the cells and incubated for 24, 48 and 72 h at 37 °C under 5% CO_2_ condition. The cell viability was determined by the SRB assay and compared to untreated cells. After cells were treated with herbal extracts, 40 µL of 50% TCA was added to each well and incubated at 4 °C for an hour. After that, plates were washed 4 times by slowing tap water. Then, the plates were completely dried and 100 µL of 0.057% (*w/v*) SRB solution was added to each well, incubated at room temperature for 30 min, and then the plate was washed with 1% (*v/v*) acetic acids 4 times to remove unbound dye. After the plate was dried, 200 µL of 10 mM Tris-based solution (pH 10.5) was added to dissolve the dye and the plate was shaken. The absorbance at 510 nm was measured by a microplate reader (BioTek, Winooski, VT, USA) [64]. The cytotoxicity of both ethanolic extracts (HCT and PR) was determined by SRB assay. Briefly, both MCF-7 and MDA-MB-231 cells were plated in a 96-well culture plate at 10,000 cells per well. After 24 h, the cells were treated with various concentrations of HCT and PR (100–500 μg/mL) separately, and incubated for 24, 48, and 72 h at 37 °C, under 5% CO_2_ condition. The cell viability was measured by SRB assay as has been described previously.

### 4.10. Apoptosis Determination by Flow Cytometry

After treatment with both ethanolic HCT and PR extracts at concentrations of IC20 and IC50 for 24 h, the cells were washed twice with PBS (centrifuged at 4 °C, 500× *g* for 5 min) and stained with 50 µL of binding buffer containing the reagent annexin V-FITC and PI for 15 min. Then, the binding buffer 250 µL was added to the stained cells and the samples were processed using a BD FACScan^™^ flow cytometer (BD Biosciences, San Jose, CA, USA). Results were analyzed and reported as the percentage of selected cells that are positive for either annexin V-FITC or PI or both [65].

### 4.11. Determination of Caspase 9, -8 and -3 Activities 

After both cells were treated with various concentrations of IC20 and IC50 of HCT and PR for 12 h, the harvested cells were briefly lysed with a lysis buffer at 4 °C. Protein concentrations were established using the Bradford method. Caspase-9, -8 and -3 activities were measured by colorimetric protease assay according to the manufacturer’s protocol [66].

### 4.12. Western Blotting

HCT and PR induced cell cycle arrest and apoptosis in both cancer cells, the expression levels of cell cycle related proteins (p21, cyclin D1, and CDK4) and apoptosis related proteins (Bcl-xl, BAX, Bid, Noxa, capsases-7, -8 and -9) were determined in treated breast cancer cell lines. After the herbal extracts treatment at IC20 and IC50 concentrations for apoptosis induction and lower IC20 concentration for the cell cycle progression study, the cancer cells were washed twice with phosphate-buffered saline (PBS). Then, the cells were centrifuged at 200× *g* for 10 min, the supernatants were removed, and cells’ pellets were lysed with a RIPA buffer containing protease inhibitors for 30 min on ice. The unbroken cells and cell debris were removed by centrifugation at 10,000× *g* for 10 min at 4 °C and the supernatants were collected. Protein concentrations were determined by the Bradford Assay Kit. The protein lysate (50 µg/lane) was separated on a 12.5% SDS-polyacrylamide gel electrophoresis (SDS-PAGE), using 100 volts for an hour in electrode buffer, and was transferred using 100 volts for 1.5 h, in transfer buffer onto nitrocellulose membrane. For blocking non-specific binding of the antibody, the membrane was blocked thoroughly with blocking solution for an hour at room temperature. It was then incubated with primary antibody in 0.3% Tween-TBS at 4 °C, overnight. Then, the membrane was washed with 0.3% Tween-TBS 5 times, for 5 min each, to remove excess and the non-specific binding antibody. Then, the membrane was incubated with goat anti-mouse IgG linked to horseradish peroxidase (HRP-conjugated goat anti-mouse IgG) or goat anti-rabbit IgG linked to horseradish peroxidase (HRP-conjugated goat anti-rabbit IgG), at a 1:10,000 dilution, in washing buffer for 2 h at room temperature. After that, the membrane was washed 3 times, for 10 min each by washing buffer (0.05% Tween-TBS). Finally, the protein–antibody complex was detected by using the SuperSignal^®^ protein detection kit (enhanced chemiluminescence, ECL), and exposed to Kodak X-Omat film (approximately 1–5 min), and band density was analyzed by ImageJ, National Institutes of Health (NIH), USA. Actin was used as a loading control protein in each treatment [67].

### 4.13. Colony Formation Assay

Briefly, non-toxic concentrations of herbal plant extract-treated cells were plated in 6-well plates at a density of 1000 cells/well in triplicate and incubated at 37 °C for 24 h. Colonies were washed with PBS and cultured in complete media at 37 °C for 8 days. Then, colonies were washed with PBS fixed in 95% ethanol and stained with 0.5% crystal violet and examined under light microscopy [68].

### 4.14. Cell Cycle Assay

Cell cycle arrest was investigated by using the flow cytometry technique. Briefly, cancer cells were seeded in a 6-well plate for 24 h and then cultured in 45 min. In each experiment, determinations were carried out in triplicate from three independent experiments [11].

### 4.15. Wound-Healing Assay

Briefly, both MCF-7 and MDA-MB-231 (1 × 10^6^ cells per well) were plated in 6-well plates and incubated overnight at 37 °C. After the cells were 100% confluent, they were scraped using a 200-pipette tip, and washed out slowly with PBS. The serum-free medium with or without HCT and PR at non-toxic concentrations (lower than IC20) were added and incubated for 24 h. Gap closure or migration of the cells was observed and captured at 0, 12, and 24 h under phase contrast microscopy [69].

### 4.16. Gelatin Zymography for MMP-2 and MMP-9 Secretions Assay

After the cells were treated with or without HCT and PR at non-toxic concentrations (lower than IC20) in serum-free medium, the supernatant was collected. Protein concentrations were determined using the Bradford method. Then, 10 µg of protein samples were mixed with a substrate gel sample buffer (13.3% SDS, 40% glycerol, 42 mM Tris-HCl (pH 6.5), 0.013% bromophenol blue, reagent grade gelatin (1 mg/mL)) and loaded onto the gel without prior boiling. Gels were washed and incubated in Zymo buffer for 24 h. After that, the gels were stained with 0.5% Coomassie blue and then de-stained using de-staining buffer (methanol: acetic acid: water, 4:1:5). Areas of protease appeared as clear bands against a dark blue background where the metalloproteases had digested the gelatin [70].

### 4.17. Transwell-Migration and Invasion Assay

Cell migration and invasion were assessed with modified Boyden chamber (Becton Dickinson Labware) assays. Briefly, cells, approximately 1.0 × 10^5^, were plated into the upper chamber of a polycarbonate Transwell filter chamber, with or without matrigel (Corning, NY, USA). After 24 h, cells that did not migrate were removed from the top side of the inserts with a cotton swab. Cells that migrated to or invaded the underside of the inserts were fixed in 4% paraformaldehyde, stained with crystal violet, and the migratory cells were counted under a light microscope. Cells were counted in 5 random fields per insert. Three independent experiments were carried out [71].

### 4.18. Statistical Analysis

All data represent the mean (± SD) of three independent experiments performed in triplicate. All statistical analyses were analyzed with GraphPad Prism 6.0 software (GraphPad Software, Inc., San Diego, CA, USA) by using one-way ANOVA with the post-hoc Tukey’s test and multiple variables. Statistical significance was considered when * *p* < 0.05, ** *p* < 0.01, *** *p* < 0.001.

## 5. Conclusions

The low concentration of HCT and PR ethanolic extracts can inhibit cell growth and proliferation of MCF-7 and MDA-MB-231 through the modulation of cyclin D1, CDK-4, and p21 expression. In addition, both extracts also inhibited the migration and invasion of breast cancer cells via MMP-2 and -9 abolishment. Moreover, HCT and PR extracts at high concentrations can induce apoptosis on breast cancer cells by stimulating caspase activity and upregulating the expression of apoptotic involving proteins. The present study may be useful in developing anti-cancer agents for breast cancer in the future.

## Figures and Tables

**Figure 1 molecules-25-01196-f001:**
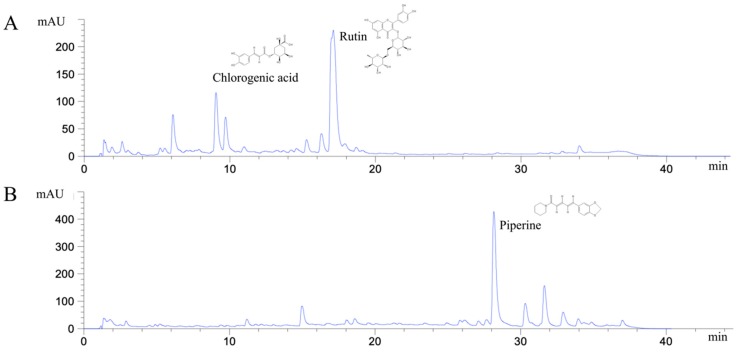
HPLC chromatograms of ethanolic *H. cordata* extract (**A**) and *P. ribesioides* extract (**B**).

**Figure 2 molecules-25-01196-f002:**
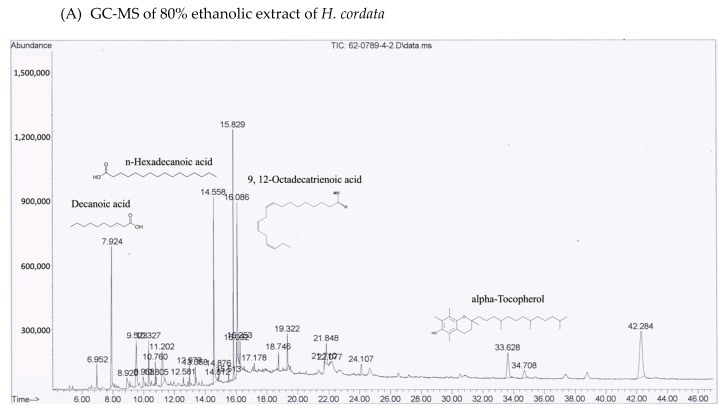
GC-MS chromatograms of both ethanolic (**A**) *H. cordata* and (**B**) *P. ribesioides* extracts. Phytochemicals in both plant extracts were analyzed using DB-5MS column with Agilent technology GC 7890A coupled to Agilent technology MSD 5975C (EI).

**Figure 3 molecules-25-01196-f003:**
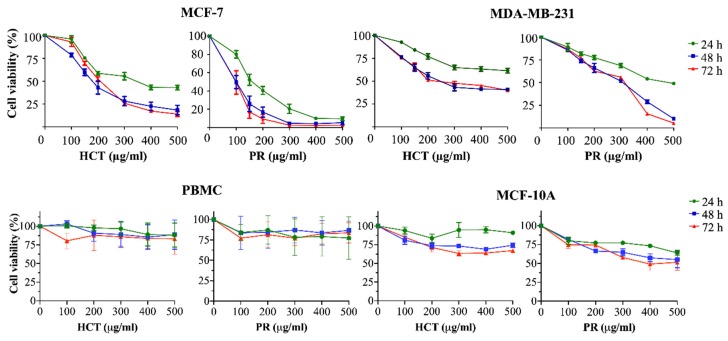
Cytotoxic effects of herbal extracts on breast cancer cells, PBMCs and MCF-10A cell line. The cells were treated with various concentrations (100–500 µg/mL) of *H. cordata* and *P. ribesioides* extracts for 24, 48 and 72 h. Cell viability was evaluated by comparing with 0.5% DMSO treated control cell, after 24, 48 and 72 h of incubation. Results are presented as mean ± SD from three independent experiments.

**Figure 4 molecules-25-01196-f004:**
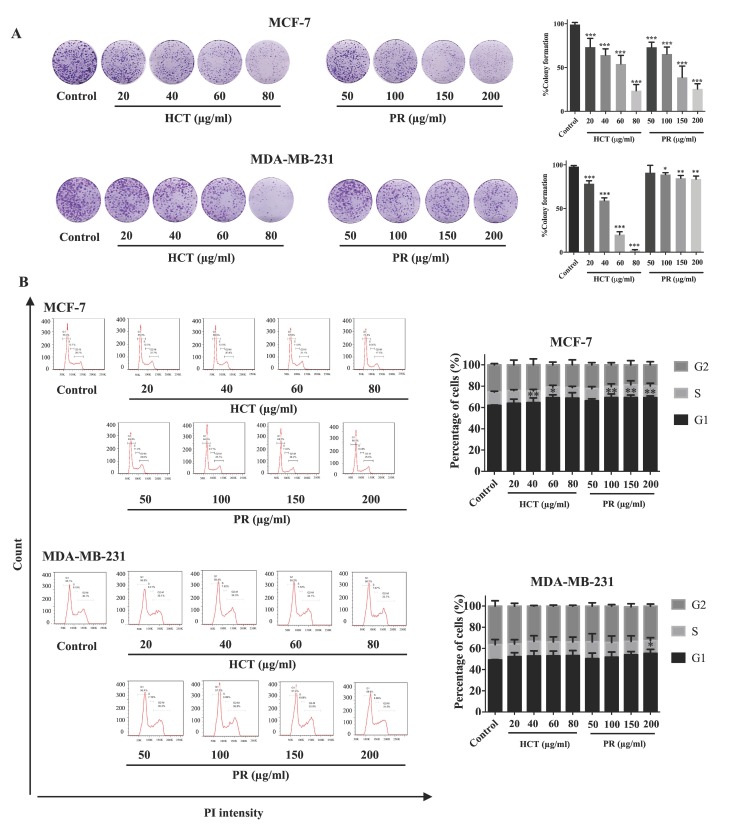
Effect of both *H. cordata* and *P. ribesioides* extracts on breast cancer cell growth inhibition and cell cycle disruption. (**A**) Colony formation assay. MCF-7 and MDA-MB-231 were treated with non-toxic concentrations of both *Houttuynia cordata* Thunb. (HCT) and *Piper ribesioides* Wall. (PR) extracts for 24 h, washed by PBS and cultured for seven days. The colonies were counted and compared with the control. (**B**) For cell cycle assay, the cells were treated with the same condition in the colony formation assay and analyzed by flow cytometry after the cells were stained with propidium iodide. Cell numbers are presented as a percentage of the total analyzed cells. Each value is presented as mean ± SD from three independent experiments. Data are the means ± SD. * *p* < 0.05, ** *p* < 0.01, and *** *p* < 0.001 vs. control.

**Figure 5 molecules-25-01196-f005:**
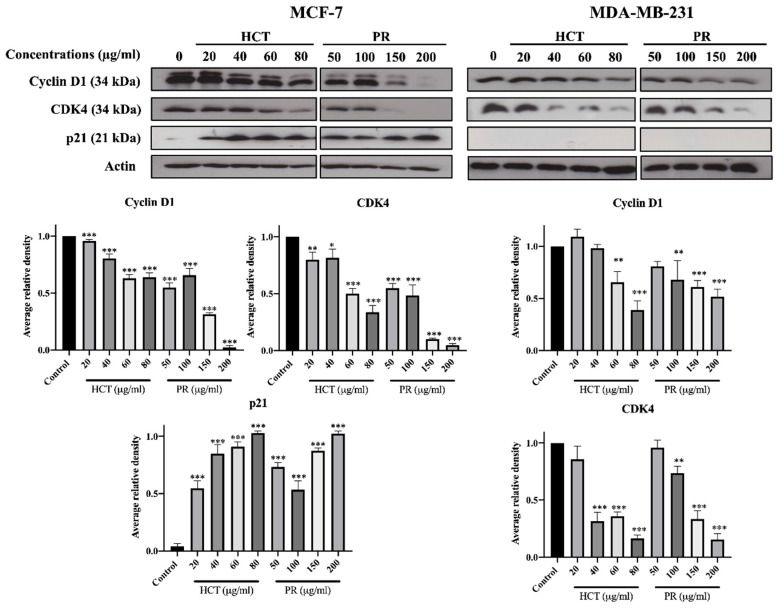
Effect of both *H. cordata* and *P. ribesioides* extracts inhibited cell cycle related protein expressions on breast cancer, by using Western blotting to investigate cyclin D1, CDK4, and p21. β-Actin acted as the loading control. Each value is presented as mean ± SD from three independent experiments. Data are the means ± SD. * *p* < 0.05, ** *p* < 0.01, and *** *p* < 0.001 vs. control.

**Figure 6 molecules-25-01196-f006:**
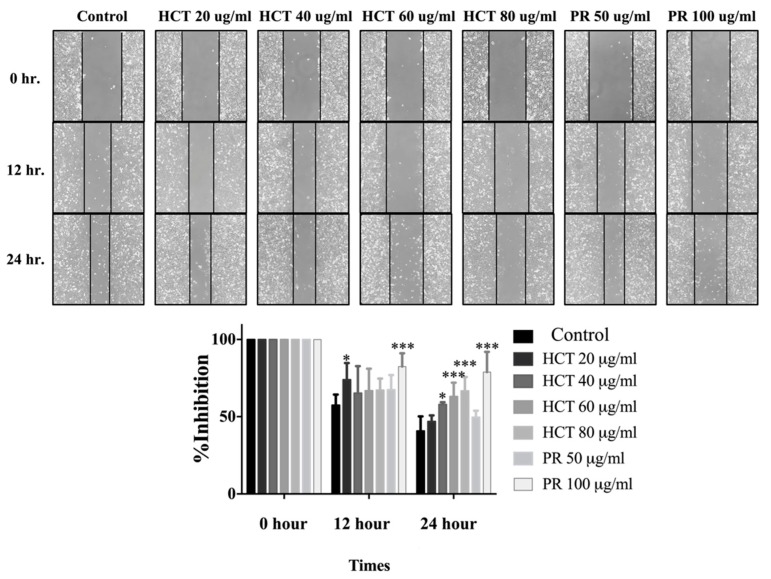
Effect of *H. cordata* and *P. ribesioides* extracts on breast cancer cell migration. Cells were treated with non-toxic doses for 24 h, and the wound healing assay was investigated. The percentage of effectiveness for inhibition on migration in bar graphs. Each value is presented as mean ± SD from three independent experiments. Data are the means ± SD. * *p* < 0.05 and *** *p* < 0.001 vs. control.

**Figure 7 molecules-25-01196-f007:**
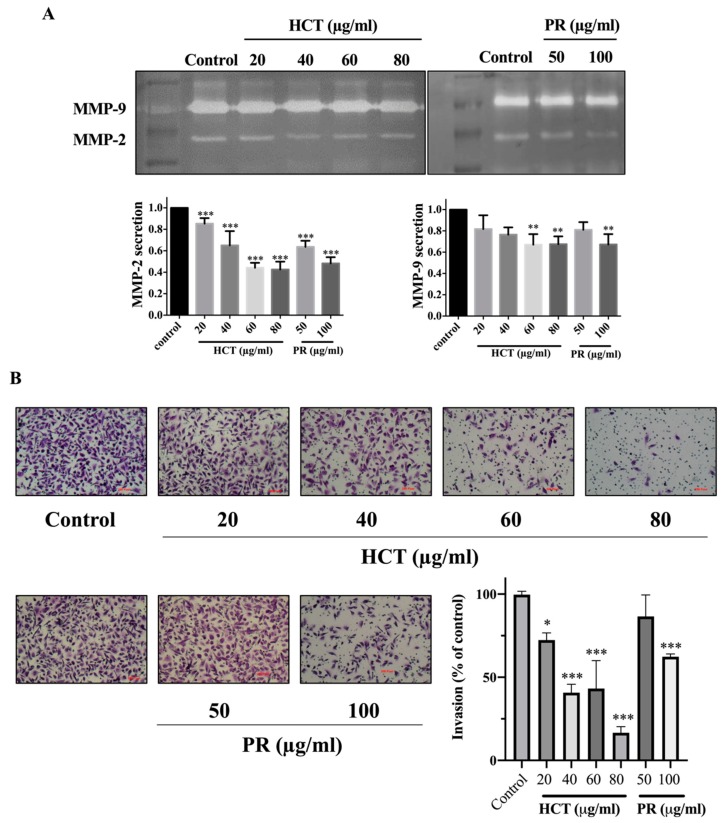
Effect of *H. cordata* and *P. ribesioides* extracts on breast cancer cell invasion. MMP-2 and -9 secretions assay (**A**) and invasion assay (**B**) were analyzed. MMP-2 and -9 were quantitated compared with the control. Each value is presented as mean ± SD from three independent experiments. Data are the means ± SD. * *p* < 0.05, ** *p* < 0.01, and *** *p* < 0.001 vs. control.

**Figure 8 molecules-25-01196-f008:**
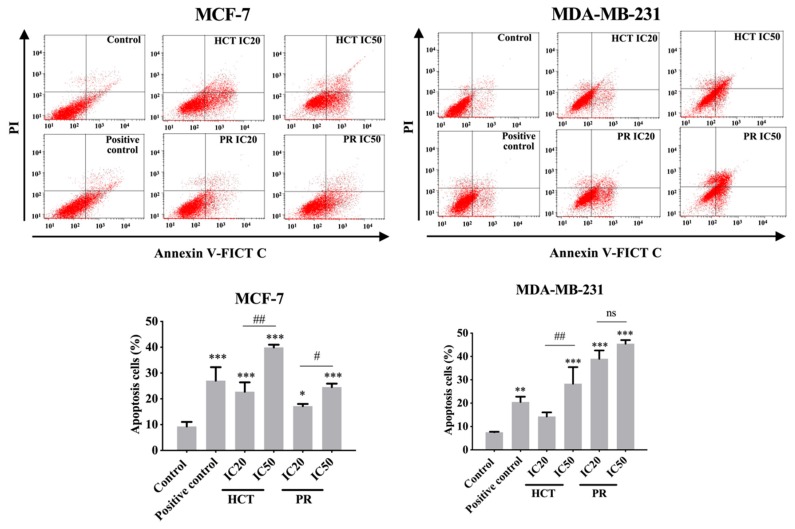
Effect of *H. cordata* and *P. ribesioides* extracts on apoptosis induction in breast cancer cells. Flow cytometric analysis of apoptosis induction on MCF-7 and MDA-MB-231 cells were treated with various concentrations of HCT and PR ethanolic extracts for 24 h. The annexin V-FITC/PI staining apoptotic cells were analyzed using flow cytometry. Camptothecin (CTP; 5 nM) was used as a positive control. Bar graphs showed the summarized data of three independent experiments from MCF-7 (left) and MDA-MB-231 cells (right), performed in duplicate compared with the untreated control. Each value is presented as mean ± SD from three independent experiments. Data are the means ± SD. * *p* < 0.05, ** *p* < 0.01, and *** *p* < 0.001 vs. control. # *p* < 0.01 and ## *p* < 0.001 vs. IC20 of each extract. ns, not significantly different.

**Figure 9 molecules-25-01196-f009:**
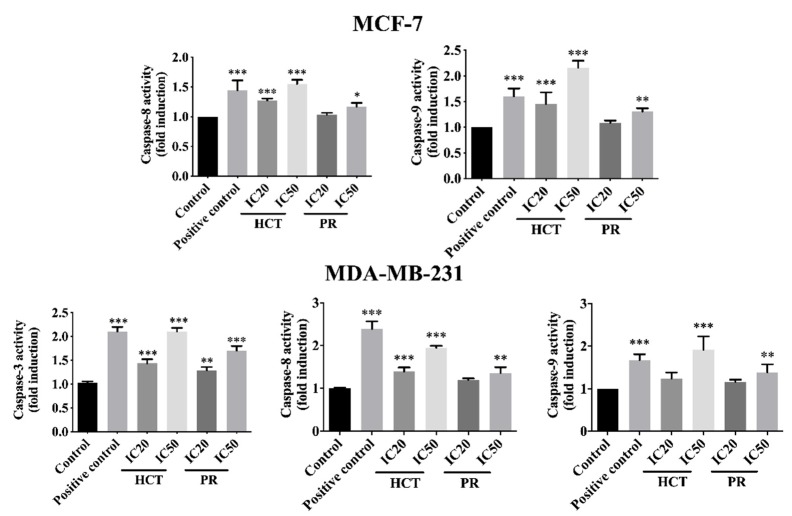
Caspases’ activities of breast cancer cells treated with *H. cordata* and *P. ribesioides* extracts. Cells were grown under the same conditions and caspases activities were analyzed, including caspase-8, -9, and -3 activities in MCF-7 (excepted caspases-3) and MDA-MB-231. Data are the means ± SD. * *p* < 0.05, ** *p* < 0.01, and *** *p* < 0.001 vs. control.

**Figure 10 molecules-25-01196-f010:**
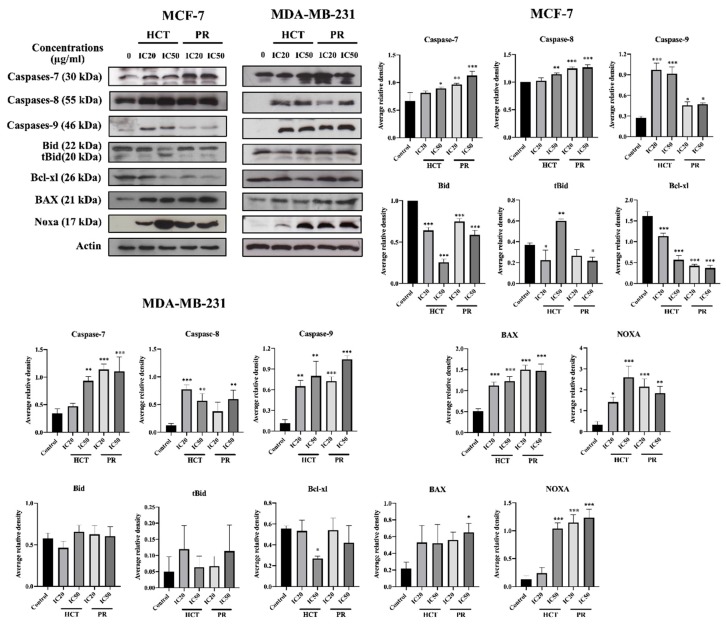
The effects of *H. cordata* and *P. ribesioides* extracts on the expression of apoptotic involving proteins in breast cancer cells. MCF-7 and MDA-MB-231 cells were treated with IC20 and IC50 of each ethanolic extract for 24 h, and whole cell lysates were prepared and subjected to Western blotting. The immunoblots were investigated for caspase-7, caspase-8, caspase-9, Bid, tBid, Bcl-xl, BAX and Noxa. β-Actin acted as the loading control. The results are normalized to the control. Data are the means ± SD. * *p* < 0.05, ** *p* < 0.01, and *** *p* < 0.001 vs. control.

**Table 1 molecules-25-01196-t001:** Determination of total phenolic and flavonoid contents and DPPH radical scavenging capacity of *H. cordata* and *P. ribesioides* extracts. Results are presented as mean ± SD from three independent experiments.

Plants	Total Phenolic Content (mg Gallic Acid/g Extract)	Total Flavonoid (mg Catechin/g Extract)	DPPH-Radical Scavenging Activity (IC50 (μg/mL))
*H. cordata*	119.8 ± 3.0	93.8 ± 1.3	234.6 ± 11.9
*P. ribesioides*	203.8 ± 10.8	95.4 ± 2.5	153.8 ± 4.4

**Table 2 molecules-25-01196-t002:** Phenolic acid compositions and flavonoids of both ethanolic *H. cordata* and *P. ribesioides*.

Retention Times (Rt)	*H. cordata*	*P. ribesioides*	Compounds
3.186	0.435 ± 0.01	0.827 ± 0.08	Gallic acid
7.908	1.64 ± 0.43	1.128 ± 0.13	Catechin
9.012	25.5 ± 3.41	n.d.	Chlorogenic acid
10.904	0.844 ± 0.14	0.543 ± 0.03	Vanilic acid
13.132	0.621 ± 0.02	1.199 ± 0.02	Ferulic acid
13.990	0.116 ± 0.02	n.d.	*p*-Courmaric acid
17.167	44.00 ± 5.61	n.d.	Rutin
18.288	1.49 ± 0.04	1.89 ± 0.01	Rosmarinic acid
21.361	0.196 ± 0.05	0.465 ± 0.05	Quercetin
22.329	n.d.	0.124 ± 0.01	Apigenin

n.d. = non-detectable.

**Table 3 molecules-25-01196-t003:** Phytochemical compounds identified in *H. cordata* by using GC-MS.

No.	Retention Time	Name of Compounds	% of Total
1	7.926	Decanoic acid	9.42
2	10.329	Dodecanoic acid	2.57
3	12.578	Tetradecanoic acid	0.54
4	12.979	Octadecane	0.9
5	14.558	*n*-Hexadecanoic acid	10.5
6	14.878	Eicosane	0.59
7	16.034	9,12-Octadecadienoic acid	1.78
8	16.091	9,12-Octadecatrienoic acid	11.67
9	16.257	Octadecanic acid	2.93
10	21.716	9,17-Octadecadienal	0.78
11	33.629	alpha-Tocopherol	5.33
12	34.711	Sesamin	2.01
13	42.287	Cholest-5-en-3-ol	14.74

**Table 4 molecules-25-01196-t004:** Phytochemical compounds identified in *P. ribesioides* by using GC-MS.

No.	Retention Time	Name of Compounds	% of Total
1	10.324	Dodecanoic acid	0.05
2	12.584	Tetradecanoic acid	0.43
3	14.581	*n*-Hexadecanoic acid	6.82
4	16.051	9,12-Octadecanoic acid	3.84
5	16.103	cis-Vaccenic acid	5.1
6	16.269	Octadecanoic acid	1.94
7	26.540	Piperine	49.51
8	42.292	β-Sitrosterol	3.46

**Table 5 molecules-25-01196-t005:** Determination of IC20 and IC50 values of *H. cordata* and *P. ribesioides* extracts of breast cancer cells, peripheral blood mononuclear cells (PBMCs) and MCF-10A cell line. All cells were treated with various concentrations (100–500 µg/mL) of *H. cordata* and *P. ribesioides* extracts for 24, 48 and 72 h. Results are presented as mean ± SD for three independent experiments.

**MCF-7**						
	***H. cordata*** **Extract (µg/mL)**			***P. ribesioides*** **Extract (µg/mL)**		
	**24 h**	**48 h**	**72 h**	**24 h**	**48 h**	**72 h**
IC20	129.3 ± 6.9	82.4 ± 7.1	129.7 ± 3.5	93.2 ± 7.2	56.4 ± 9.7	66.4 ± 12.2
IC50	347.4 ± 10.8	187.9 ± 18.2	210.8 ± 3.1	167.0 ± 11.0	99.5 ± 12.0	98.4 ± 14.1
**MDA-MB-231**						
	**24 h**	**48 h**	**72 h**	**24 h**	**48 h**	**72 h**
IC20	184.6 ± 14.0	66.9 ± 5.3	64.8 ± 3.8	174.6 ± 13.8	146.3 ± 5.6	149.5 ± 8.5
IC50	667.8 ± 11.7	279.6 ± 4.4	290.7 ± 11.9	484.5 ± 14.3	264.1 ± 12.2	254.1 ± 8.0
**PBMCs**						
	**24 h**	**48 h**	**72 h**	**24 h**	**48 h**	**72 h**
IC20	>1000 µg/mL					
IC50	>1000 µg/mL					
**MCF-10A**						
	**24 h**	**48 h**	**72 h**	**24 h**	**48 h**	**72 h**
IC20	514.6 ± 5.3	372.3 ± 9.1	275.9 ± 2.1	353.1 ± 4.5	256.3 ± 6.5	264.5 ± 2.5
IC50	>1000 µg/mL	893.8 ± 8.4	679.5 ± 7.9	879.5 ± 2.6	523.0 ± 1.7	442.9 ± 1.0

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
