# Peer review of "The Effects of Houttuynia cordata Thunb and Piper ribesioides Wall Extracts on Breast Carcinoma Cell Proliferation, Migration, Invasion and Apoptosis"

_molecules, 2020, doi:10.3390/molecules25051196_

Round 1
Reviewer 1 Report
This manuscript investigated the natural products isolated from the herbs (HCT and PR) that contributed to anti-proliferative and pro-apoptosis properties. The author fist analyzed the ethanol extracts of the herbs using LC-MS and GC-MS, then applied the crude extracts to a variety of biological assays to determine their anti-cancer activities. However, the biological activities of major compounds in the crude extracts have already been reported and the authors have also included in their discussion, therefore, the novelty of the manuscript is questionable.
Specific comments:
(1) Please show chemical structures of the identified compounds in Fig. 1 and Table 1.
(2) Please determine the specific compounds that contribute to the bioactivities. Conclusions resulted from crude extracts are not informative.
(3) Please determine synergistic effects of compounds in the crude extracts.
(4) What are the activities of the compounds in the extracts towards normal mammalian cells? What are the specificities of the extracts towards cancer cells?
Reviewer 2 Report
The current manuscript of Subhawa et al., entitled as “The effects of Houttuynia cordata Thunb and Piper ribesioides Wall extracts on breast carcinoma cell proliferation, metastasis and apoptosis” presents the antiproliferative effect of two plant extracts.
However, numerous new anti-cancer drugs have been developed in the last couple of decades, the demand for a new class of anti-cancer agents is still high, due to the increasing number of drug-resistance cases, serious side-effect, etc. One of the most promising sources of new antiproliferative medicine is plant kingdom (FDA data showed that 40% of the approved molecules are natural compounds or inspired by them, from which, 74% are used in anticancer therapy). Therefore the significance of the current study is high. The antiproliferative effect of the extract of Houttuynia cordata is already know, moreover several publications reported its possible mechanism of action (Anticancer Res. 2017. 37:6619-6628; Integr Cancer Ther. 2017. 16:360-372; J Biomed Sci. 2013. 20:18). Therefore the former research on H. cordata significantly reduces the novelty of the current manuscript.
Major concerns:
- The authors used untreated cells as control. Since the untreated cells are naïve, they are not the appropriate negative control. The appropriate control would have been vehicle-treated cells. Using vehicle-treated control can guarantee that the effect of vehicle is excluded. Moreover the authors did not indicate what kind of vehicle they used to dilute the extracts.
- All the figures need to be improved. Several axis legends are illegible.
- The quantification and the statistical analysis of Western immunoblots are completely missing. However in the method section the authors mentioned that they used ImageJ to analyze band density, but the results of the analysis are not shown.
- I would highly recommend to rewrite the abstract section due to grammatical errors like in lane 13: “However, in breast cancer, the mechanisms responsible for the inhibition of cancer invasion are yet to be clarified.” (The correct sentence would be: However, in breast cancer, the mechanisms responsible for the inhibition of cancer invasion have not been clarified yet.”) Lane 17: “At low concentrations, it demonstrated that colony formation was decelerated and cell cycle arrest at G1 phase.” At low concentration of which extract or both extracts?
Questions and remarks:
Introduction:
Lane 68: “This study may be useful in the development of Thai medicinal herbs as food supplements based on these plants to effectively protect people from cancer in order to prolong life among the elderly.”….” consumed on a daily basis” I strongly recommend that the authors should reconsider these statements. If an extract shows remarkable anti-cancer activity, its consumption as food supplement (especially on daily basis) is not recommended at all (possible side-effect, possible drug-interaction..etc). The compounds responsible for the activity should be identified, tested in animal models, and the research should follow the phases of the classical drug development.
Materials and Methods:
- The authors did not specify which part of the plants (like whole plant, aerial, roots..etc) were used for extraction.
- Lane 412: “The total phenolic compounds in these two plant extracts were determined by using Folin-Ciocalteu assay.” Lane 441: “The total phenolic compounds were be calculated by HPLC peak area under the curve compared with standard calibration curve.” It is not clear how the authors determined the total phenolic compounds, since the two statements are contradictory. Certain phenolic constituents can be identified by HPLC, but the authors should clarify how they identified chlorogenic acid, rutin and piperine in the extracts by HPLC (Fig 1 A and B).
- Lane 461: “9 Cell viability and proliferation assay” the two paragraph in this section should be combined.
Results:
- Lane 156. : “The results show that cyclin D1 and CDK4 decreased in both MCF-7 and MDA-MB-231 cells…” There is no quantification for Western results (Fig. 4E), therefore no statistical analysis was performed to compare the effect of the extracts to the control. It means the author`s conclusion about cyclin D1 and CDK4 is speculative.
- The quality of images in Fig 4A is really low.
- Figure 4C is the quantification for Fig 4A. They should be presented together.
- Fig 4D is the quantification for Fig 4B. I suggest to show them together.
- Lane 173: “4 Effect of H. cordata and P. ribesioides on Breast cancer progression and metastasis” The authors investigated the effect of the extracts on cancer invasion and cell migration. Those characteristics are obviously associated with cancer progression and metastasis, but I would suggest to modify the title of this section, since cancer progression and the formation of metastasis are very complex and influenced by several other factors. Moreover the authors also used “metastasis” in the main title of the manuscript. Since they did not test the effect of the extract in an animal model, which would give more insight how they can really affect cancer progression and metastasis, I would not recommend to use this word in the main title. (“Anti-metastatic activity of HCT and PR” is used other places in the text, please reconsider it.)
- The quality of images in Fig 5B and C is really low.
- The authors should present representative images and the corresponding quantification together (like Fig 5: A+D, B+E, C+F).
- The expression level of MMP-9 and -2 on the zymogram pictures (Fig 5A) do not correspond well with the numbers on the graph (Fig 5D). Minimal differences between treated and control can be visible on the images.
- It is not clear how the authors quantified the images of wound healing assay (Fig B and E). As far as I know cell analyzer software coupled with complete imaging system can be used for that purpose.
- Lane 213: “The percentage of early apoptotic and late apoptotic cells were significantly increased in both cancer cells in concentration-dependent manners (Figures 6A, 6B).” the authors did not perform statistical analysis to compare the effect of IC20 and IC50 doses. That`s why the concentration-dependent manner is a speculative statement.
- There is no quantification and statistical analysis for Western immunoblotting in Fig 7C. However the authors claimed in the text that the extracts could significantly and dose-dependently influence the expression of pro-and anti-apoptotic proteins, which is not fully proved without appropriate quantification and statistical analysis.
- The authors did not indicate the molecular weight of the proteins in any immunoblots.
- Lane 357: “The present study shows that both ethanolic HCT and PR extracts induced apoptosis via caspase-dependent pathways.” This statement is unproven, the authors should conduct an experiment with caspase-inhibitors. If the caspase-inhibitors can diminish the effect of the plant extract, the caspase-dependent activity of the extracts would be fully confirmed.
- Lane 286: “Therefore, we assumed that these substances that had been found in both ethanolic plant extracts seem to be an active compound responsible for their bioactivities.” Without appropriate results, like bioactivity-guided fractionation, the active compounds responsible for the biological activity are unknown. However, some of the molecules identified in the extracts, like cholorgenic acid, rutin, and piperine possess anti-cancer effect, a report identified bioactive alkaloids with anti-proliferative potentials in the extract of cordata (Arch Pharm Res. 2001. 24:518–21). How these compounds contribute the overall effect of the extract, how they can interact (synergistic, antagonistic or no effect), and which compound is the major one responsible for the effect (sometimes not the main compound shows the highest biological activity) remain unidentified.
Further suggestions:
- The authors should check spelling mistakes and typo errors in the entire document.
- Moderate English changes required due to some grammatical errors.
- The reference list should follow the instruction of the journal (like abbreviated journal name).
Regarding my concerns and cavity of the analysis of the results, I would recommend major revision of the manuscript.
Reviewer 3 Report
The paper of Subhawat Subhawa et al. represents an attempt to elucidate the mechanisms of the anti-proliferative and pro-apoptotic effects of Houttuynia cordata and Piper ribesioides extracts in breast cancer cell model. Although its topic is interesting, the paper is at a rather preliminary stage and it lacks the mechanistic aspect. Therefore, in its present shape, it cannot be recommended for publication in Molecules. In particular:
- HC extract content and activity has already been studied in breast cancer model. Therefore, the rational for the studies on HC extract activity is unclear. Moreover, they claim that the extracts inhibit metastasis but no in vivo studies were performed;
- estimated IC50 values for both extracts are relatively high. Considering the limited bioavailability of their phenolic compounds, is it possible to get corresponding concentrations of their compounds in the breast cancer microenvironment in vivo?;
- considering the lack of D1 cyclin in MCF cells exposed to PR at 200 ug/ml, relatively high fraction of S/G2 cells is somewhat surprising;
- Fig. 5: effect of the extracts on wound healing/invasion efficiency can easily be ascribed to its inhibitory effect on cell proliferation. Single cell trajectories should be traced to clarify this point;
- the Authors claim that the they scrutinized the mechanisms of the effect of HC and PR extracts on breast cancer cells, whereas they show only cell reactions to these extracts, without getting insight into the processes that induce these reactions. Accordingly, they should perform more detailed analyses of signaling pathways involved in these reactions;
- Comparative analyses of the activity of purified flavonoids and phenolic acids detected in the extracts could be considered by the Authors to increase the impact of the story;
-Discussion should be rewritten. The novel aspects of the story should be stressed and clearly distinguished from already published data;
- the clarity of the message suffers from chaotic structure and numerous syntax and lexical errors. For instance, see 1st phrase of Discussion. Accordingly, the paper should be reviewed by native speaker with the experience in writing scientific texts.
Round 2
Reviewer 1 Report
The authors have addressed some of the questions, however, the answer to question 3 is not satisfying.
Reviewer 2 Report
Please delete ratio numbers (like cyclin D1/actin, CDK4/actin etc..) from Western images. The graphs represent those numbers.
Reviewer 3 Report
I have no more comments